# Benzene and NO_2_ Exposure during Pregnancy and Preterm Birth in Two Philadelphia Hospitals, 2013–2017

**DOI:** 10.3390/ijerph191610365

**Published:** 2022-08-19

**Authors:** Kathleen M. Escoto, Anne M. Mullin, Rachel Ledyard, Elizabeth Rovit, Nancy Yang, Sheila Tripathy, Heather H. Burris, Jane E. Clougherty

**Affiliations:** 1Department of Epidemiology and Biostatistics, Dornsife School of Public Health, Drexel University, Philadelphia, PA 19104, USA; 2School of Medicine, Tufts University, Boston, MA 02111, USA; 3Division of Neonatology, Children’s Hospital of Philadelphia, Philadelphia, PA 19104, USA; 4Center for Public Health Initiatives, Perelman School of Medicine, University of Pennsylvania, Philadelphia, PA 19104, USA; 5Department of Environmental and Occupational Health, Dornsife School of Public Health, Drexel University, Philadelphia, PA 19104, USA; 6Department of Environmental Health, Harvard TH Chan School of Public Health, Cambridge, MA 02115, USA; 7Department of Pediatrics, Perelman School of Medicine, University of Pennsylvania, Philadelphia, PA 19104, USA; 8Leonard Davis Institute, University of Pennsylvania, Philadelphia, PA 19104, USA

**Keywords:** NO_2_, nitrogen dioxide, benzene, preterm birth, Philadelphia, spontaneous preterm birth, medically indicated preterm birth

## Abstract

Infants born preterm are at risk of neonatal morbidity and mortality. Preterm birth (PTB) can be categorized as either spontaneous (sPTB) or medically indicated (mPTB), resulting from distinct pathophysiologic processes such as preterm labor or preeclampsia, respectively. A growing body of literature has demonstrated the impacts of nitrogen dioxide (NO_2_) and benzene exposure on PTB, though few studies have investigated how these associations may differ by PTB subtype. We investigated the associations of NO_2_ and benzene exposure with sPTB and mPTB among 18,616 singleton live births at two Philadelphia hospitals between 2013 and 2017. Residential NO_2_ exposure was estimated using a land use regression model and averaged over the patient’s full pregnancy. Benzene exposure was estimated at the census tract level using National Air Toxics Assessment (NATA) exposure data from 2014. We used logistic mixed-effects models to calculate odds ratios for overall PTB, sPTB, and mPTB separately, adjusting for patient- and tract-level confounders. Given the known racial segregation and PTB disparities in Philadelphia, we also examined race-stratified models. Counter to the hypothesis, neither NO_2_ nor benzene exposure differed by race, and neither were significantly associated with PTB or PTB subtypes. As such, these pollutants do not appear to explain the racial disparities in PTB in this setting.

## 1. Introduction

Preterm birth (PTB) occurs in one in ten live births in the United States [1]. A major risk factor for infant mortality and future adverse health outcomes [2,3], PTB also occurs 1.5 times more often among Black individuals compared to White individuals [4], likely due to social, environmental and structural factors that continue to systemically differ by race [5]. There are two substantively different subtypes of PTB. Spontaneous PTB (sPTB) occurs after preterm labor or the premature rupture of membranes prior to 37 weeks of gestation. Medically indicated PTB (mPTB) is provider-initiated due to concern for maternal or fetal wellbeing from conditions such as preeclampsia or poor fetal growth prior to 37 weeks of gestation. Though certain risk factors for each subtype are shown to differ (young maternal age is more strongly associated with sPTB, whereas obesity and hypertension are more strongly associated with mPTB) [6,7,8,9,10], few studies have considered how environmental exposure to factors such as air pollution may contribute differently to each PTB phenotype.

There is a growing body of literature demonstrating the impacts of NO_2_ and benzene exposure during pregnancy on adverse birth outcomes [11,12,13]. NO_2_ is primarily emitted from motor vehicles and power plants and is a ubiquitous, though varying, form of exposure in urban environments. Benzene, a volatile organic compound (VOC), is a carcinogen emitted from industrial sources such as oil refineries. Benzene was of particular interest in this setting due to a large oil refinery operating in South Philadelphia during this period, and several large oil refineries operating in the region. NO_2_ exposure during pregnancy has been associated with adverse birth outcomes, including PTB [12,13], though some studies remain inconclusive [14,15]. While less studied, benzene exposure has also been associated with PTB [11,12]. The biological mechanism through which NO_2_, benzene and other air pollutants may lead to adverse birth outcomes is not fully understood, but may involve oxidative stress and impaired oxygen transport to the fetus [16,17,18].

This study aimed to examine the associations of NO_2_ and benzene exposure during pregnancy with PTB phenotypes in a cohort of singleton births at two Philadelphia hospitals between 2013 and 2017, after controlling for other characteristics. Due to ongoing residential segregation, which may co-vary with both pollutant exposure and PTB rates, we also explored race-stratified analyses.

## 2. Materials and Methods

### 2.1. Study Population

We used a subset of the *GeoBirth* cohort, which includes all live singleton births at the Hospital of the University of Pennsylvania and Pennsylvania Hospital in Philadelphia, PA from 23 April 2013 to 4 March 2017 (*n* = 24,683 total) [19,20], for which we had curated electronic health records and air pollution data. We retained the 19,471 patients living in Philadelphia County for whom pollution exposure assignments were available. Infant records with unrealistic birth weights (<200 g or >7000 g) or who were at <20 weeks’ or >45 weeks’ gestation at birth were excluded from the analysis (*n* = 302). Additionally, patients (*n* = 513) missing covariate information were excluded, except for body mass index (BMI) (18%); given the high prevalence of missing BMI data, we created a categorical variable with a ‘missing’ category to retain these births in the analysis (Figure 1).

### 2.2. Exposure Estimates

NO_2_ exposure was estimated using the average predicted NO_2_ concentration within a 300 m buffer around each patient’s home on the date of delivery, averaged over each patient’s entire pregnancy. We used a NO_2_ surface developed using land use regression (LUR) modeling, based on spatial source covariates derived in ArcGIS and monitoring data collected in January 2018 at 48 sites distributed across Philadelphia County, plus 13 spatially representative sites monitored throughout 2018. After testing a large suite of GIS-based source covariates—including traffic density indicators, roadway descriptors, land use/built environment metrics, industrial emissions and transportation facilities—the final LUR covariates included impervious surface within 300 m of the sampling site, distance to the nearest bus stop, distance to a navigable river and kernel traffic density within 300 m (R^2^ = 0.77). The final LUR surface was produced by predicting the NO_2_ concentration at each centroid of a 30 m × 30 m grid.

The NO_2_ estimates were temporally adjusted to the time period matching each patient’s pregnancy using Equation (1), below, and daily NO_2_ data from a Camden, New Jersey EPA Air Quality System (AQS) monitor, directly downwind of the Philadelphia area, and by providing more consistent data availability than did the Philadelphia AQS monitors for the period 2010–2018.
NO_2_ [loc x_t = 2_] = loc x_t = 1_ − AQS_t = 1_ + AQS_t = 2_(1)

The predicted NO_2_ exposure around a patient’s home during pregnancy (NO_2_ (loc x_t = 2_)) was calculated by averaging the LUR surface grid cell centroids within 300 m of each home, and was temporally adjusted by subtracting the mean NO_2_ at the Camden monitor during the two-week air monitoring period (which provided the spatial data for the LUR model) in January 2018 (AQS_t = 1_ = 19.46 ppb) from the LUR-predicted NO_2_ at the 300 m buffer around a patient’s home in January 2018 (loc x_t = 1_), replacing it with the mean NO_2_ at the Camden monitor during a specific patient’s pregnancy (AQS_t = 2_). Each patient’s period of pregnancy was estimated as the date of the last menstrual period (back-calculated using the best obstetric estimate of gestational age) until the date of birth. Trimester-specific NO_2_ exposure was calculated using the mean NO_2_ exposure in the first 12 weeks for the first trimester, the mean NO_2_ exposure between weeks 13 and 26 for the second trimester and the mean NO_2_ exposure between week 27 and the date of birth for the third trimester. Only patients that delivered between 30 and <37 weeks were included in the third trimester cohort.

Benzene exposure was calculated using census tract-level National Air Toxics Assessment (NATA) exposure data from 2014, the latest NATA estimates available, and the only NATA year during our follow-up [21]. The NATA exposure estimates were tract-level benzene concentrations, combined with census data, climate data and human activity patterns, to estimate the average exposure for a person living in that tract [22]. We assigned NATA estimates based on the patient’s census tract of residence. As Philadelphia is relatively dense, the patient census tracts were generally very small (mean area = 0.86 km^2^ (SD = 0.93 km^2^)), suggesting better accuracy than in less dense regions of larger census tracts.

### 2.3. Birth Outcomes and Covariate Ascertainment

The main outcome of interest was PTB, defined as birth before 37 weeks of gestation, with further distinction between spontaneous versus medically indicated PTB. sPTB was defined as birth following preterm labor or the spontaneous rupture of membranes. mPTB was defined as birth after labor induction or cesarian birth performed due to concern for maternal or fetal wellbeing. Each PTB was independently adjudicated by two blinded reviewers. Where there was disagreement, the chart went to a third senior reviewer for final assignment of the PTB phenotype. The covariates used in the models that may be associated with PTB included maternal race/ethnicity (Asian, Black non-Hispanic, Hispanic, other/unknown, White non-Hispanic), maternal age (<25, 25–34, ≥35 years), BMI (<25, 25–<30, ≥30 kg/m^2^, missing), insurance status (private, public/Medicaid/other), nulliparity (yes, no) and census tract-level percentage of poverty.

### 2.4. Analysis

To estimate the associations of benzene and NO_2_ exposure during pregnancy with PTB outcomes, we used logistic mixed-effects models with the birth hospital and the patient’s census tract as random intercepts to account for clustering. NO_2_ and benzene exposure were represented in parts per billion (ppb) and the models used standard deviation increments as the independent variables. Both pollutants were included in the models. We compared the unadjusted and adjusted associations of NO_2_ and benzene with overall PTB, and with sPTB and mPTB separately. We also re-ran all the models stratified by race. Statistical analyses were run using R (Version 3.6.1, R Core Team: Vienna, Austria) [23] on RStudio (Version 1.2.5001, RStudio, PBC: Boston, MA, USA) [24]. Demographic and exposure data were mapped at the census tract level using qGIS3 (Version 3.18.3, QGIS Development Team) [25].

## 3. Results

The demographic characteristics and air pollution estimates for the cohort are shown in Table 1. Of the 19,169 births, 1708 (8.9%) were PTB (*n* = 1035 (5.4%) sPTB and *n* = 657 (3.4%) mPTB). There were 16 PTBs that were not able to be classified as either sPTB or mPTB. Patients with PTB were more likely to self-identify as Black and be publicly insured. With respect to characteristics and PTB phenotypes, patients with mPTB (but not sPTB) were older and had higher BMIs. The mean NO_2_ and benzene exposure levels were similar among patients with term birth and PTB. The mean NO_2_ and benzene exposure levels were similar among patients with sPTB and mPTB.

Benzene exposure estimates by census tract are mapped in Figure 2a. The benzene exposure map shows the highest exposure concentrations in west South Philadelphia and parts of North Philadelphia. The exposure decreases as the distance from these areas increases. The spatial component of the NO_2_ LUR model shown in Figure 2b demonstrates higher exposure in Center City and South Philadelphia, as well as near highways and major roads.

Figure 3 shows the spatial distribution of the study cohort patients from the two hospitals. The majority of the patients at the Hospital of the University of Pennsylvania lived in West and Southwest Philadelphia (Figure 3a). The majority of the patients at Pennsylvania Hospital lived in Center City and South Philadelphia (Figure 3b). Figure 4 shows the racial distribution of the patients in the birth cohort. Black patients primarily lived in West and Southwest Philadelphia, in addition to west South Philadelphia, close to the oil refinery, while White patients lived in Center City and other parts of South Philadelphia.

In the unadjusted models, we did not detect associations of benzene or NO_2_ with overall PTB (Table 2). In the adjusted models, we also did not detect associations of benzene or NO_2_ with overall PTB. When stratified by race, we also did not observe significant associations of benzene or NO_2_ and with overall PTB. With respect to associations with sPTB, in the unadjusted models, we observed no significant associations with benzene or NO_2_. In the adjusted models, we also did not detect significant associations of benzene or NO_2_ with sPTB (Table 2). Race-stratified sPTB models also showed no association with benzene or NO_2_. With respect to mPTB, the unadjusted analyses showed negative associations of benzene and NO_2_ with mPTB, but adjustment nullified the association. Models stratified by race also showed no association of benzene or NO_2_ with mPTB.

The trimester-specific NO_2_ exposure models with benzene showed a negative association with mPTB in the unadjusted NO_2_ model that was nullified by adjustment, and all other trimester-specific adjusted models showed no association with overall PTB, sPTB or mPTB (Appendix A). The unadjusted and adjusted models stratified by hospital also showed no association with overall PTB, sPTB or mPTB (Appendix B Table A1).

## 4. Discussion

Our study did not find evidence that exposure to benzene and NO_2_ during pregnancy was associated with PTB in a cohort of patients from two hospitals in Philadelphia. Adjusting for patient and census tract-level characteristics and including hospital and census tract as random intercepts nullified the negative unadjusted associations of benzene and NO_2_ with mPTB. When stratified by race, we also found no association of benzene and NO_2_ with PTB or PTB phenotypes. This suggests that other types of environmental exposure may be more likely candidates in explaining the racial disparities in PTB in Philadelphia, in addition to other health and socioeconomic factors.

The lack of association may have also been driven by other factors that we could not account for in the models. While we hypothesized that NO_2_ and benzene may be toxic to pregnancies, it is likely that other factors have stronger effects. Certain individual- and health-level factors are more predictive of preterm birth outcomes. For example, high BMI and smoking are associated with preterm birth outcomes [26,27], in addition to health factors such as diabetes mellitus and high blood pressure [28,29]. However, we consciously chose not to adjust for variables that may be on the causal pathway between pollution and PTB, such as hypertension (Appendix A).

These results are consistent with other studies, such as an NYC multi-center birth study from 2008 to 2010 that explored the relationship between LUR-predicted exposure to PM_2.5_ and NO_2_ and sPTB but found negative associations due to confounding by hospital characteristics [14]. A meta-analysis of several pollutant and birth outcome studies found a pooled OR for the association between NO_2_ and PTB that was not statistically significant [30]. Our study differs from other studies’ findings, including those from a cohort of patients in Valencia, Spain from 2003 to 2005 that used LUR models to predict benzene and NO_2_ exposure and found a positive association between exposure to these pollutants and the risk of PTB [12]. In addition, a large study from Shanghai that examined the effects of LUR-predicted NO_2_ and PTB found a positive association between third-trimester NO_2_ exposure and PTB [31].

Multicenter studies such as this one are important for understanding the effect of environmental exposure on adverse birth outcomes because they provide a more representative sample of patients that give birth in an area. As shown in Figure 3, the spatial distribution of patients’ addresses between the two hospitals in the birth cohort differ substantially. While the cohort had data from only two of the five birth hospitals in Philadelphia, the births at these hospitals together made up 47% of all Philadelphia births from 2013 to 2017 [32]. Although stratifying the results by hospital did not reveal associations between the pollutants and PTB outcomes (Figure 4), including the hospital as a random intercept in the final models reduced the between-hospital variability.

There were several factors that may have contributed to the lack of association seen in the study. Reliable smoking data were unavailable, though smoking affects health overall and impacts benzene and NO_2_ exposure during pregnancy, as both are present in cigarette smoke [33]. However, only 7.2% of pregnant people in 2016 reported smoking during pregnancy [34]. Alcohol consumption and drug use data were also unavailable. In addition, many White patients lived in the areas of Philadelphia, such as Center City, with the highest traffic emissions (Figure 4), but also had higher average socioeconomic status and better access to healthcare. While we had access to insurance data, we did not have other individual-level socioeconomic variables. It is also possible that buffering factors such as air conditioning in wealthier areas with high pollution could have reduced exposure to NO_2_ and benzene in these areas. The spatial distribution of wealthier White patients in areas with higher pollution may mask the possible effects of benzene and NO_2_ on PTB. Finally, benzene air monitoring data from 2014 was at the census tract level and relied on the average exposure in 2014. Thus, in contrast to the NO_2_ estimates, the benzene estimates were not specific to each patient’s pregnancy period, which could have diluted associations that might have otherwise been detected during exposure peaks. Finally, we did not have information on the patients’ work addresses, and it is possible that patients moved residences during pregnancy, which may have led to exposure misclassification.

## 5. Conclusions

In conclusion, an analysis of 18,616 births in Philadelphia did not reveal associations of ambient NO_2_ or benzene exposure with PTB or its phenotypes, nor did it support the hypothesis that these exposures might partially explain the racial disparities in these outcomes. Our findings suggest that other types of exposure are likely responsible for the ongoing racial disparities in PTB in Philadelphia, while controlling for other socioeconomic factors. Ultimately, more thorough and consistent air monitoring of benzene, NO_2_ and other air pollutants may improve our understanding of the relationships between air pollution and PTB and may provide more insight into the environmental factors that affect PTB disparities.

## Figures and Tables

**Figure 1 ijerph-19-10365-f001:**
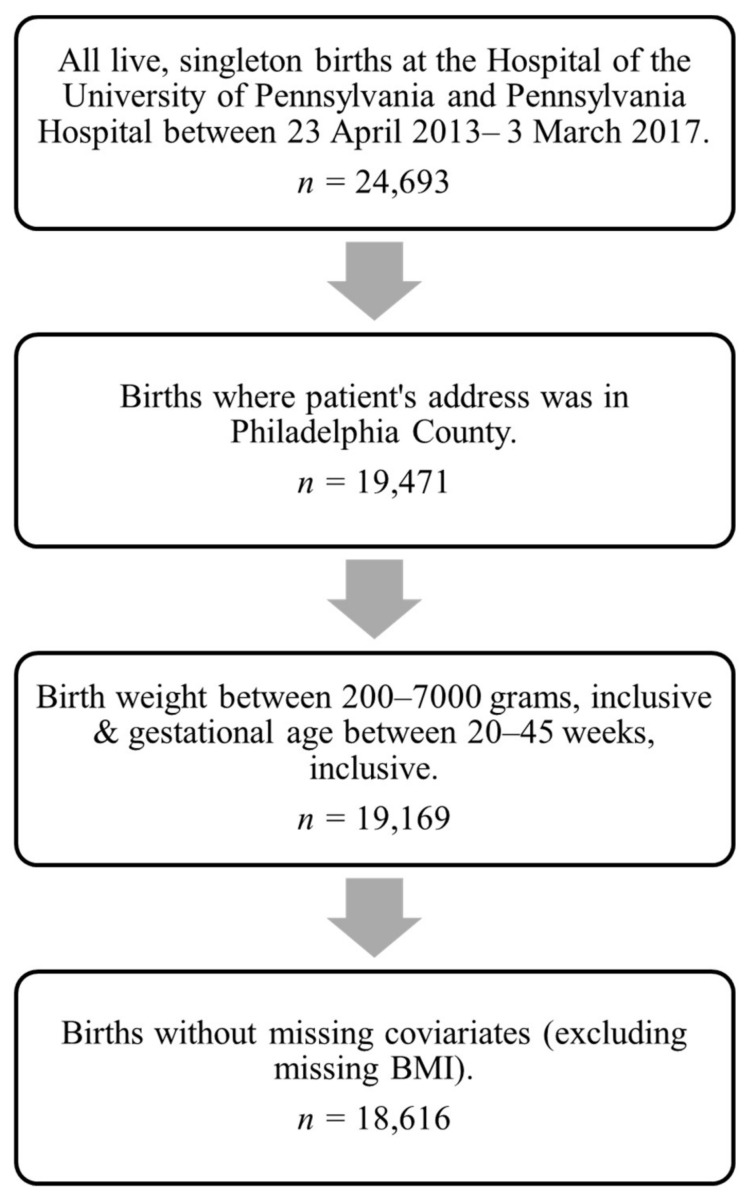
Analytic cohort development.

**Figure 2 ijerph-19-10365-f002:**
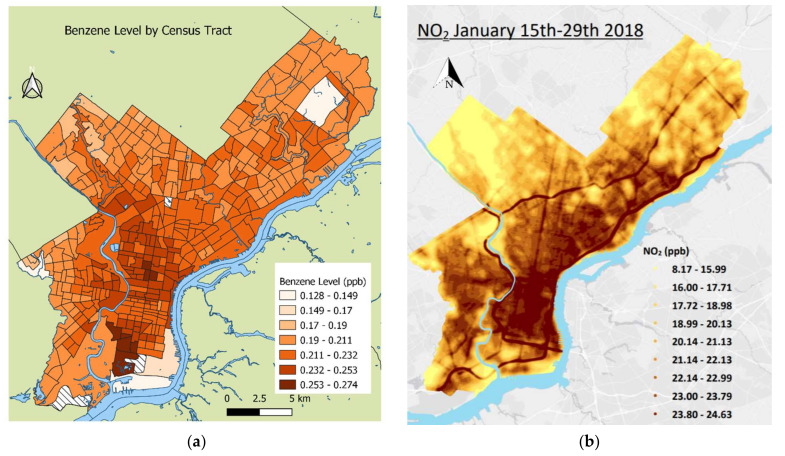
(**a**) Benzene level by census tract in Philadelphia County; (**b**) spatial component of LUR model showing NO_2_. This map does not account for temporal component, which is individualized to the patient.

**Figure 3 ijerph-19-10365-f003:**
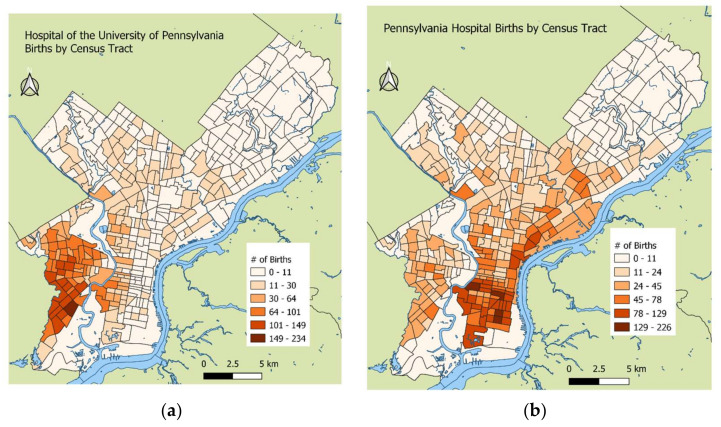
Number of births per census tract by hospital: (**a**) Hospital of the University of Pennsylvania; (**b**) Pennsylvania Hospital.

**Figure 4 ijerph-19-10365-f004:**
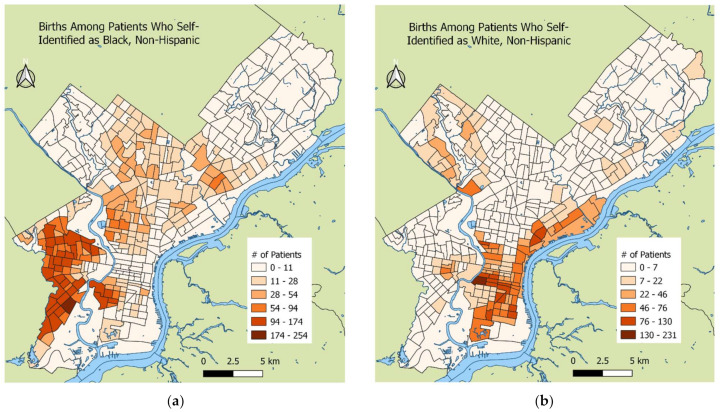
Maternal race by census tract. (**a**) Patients who self-identified as Black, Non-Hispanic; (**b**) Patients who self-identified as White Non-Hispanic.

**Table 1 ijerph-19-10365-t001:** Maternal demographics, environmental characteristics and birth outcomes of the study cohort, stratified by term birth, preterm birth (PTB), spontaneous PTB (sPTB) and medically indicated PTB (mPTB).

	Overall	Term	PTB	sPTB	mPTB
**Characteristics**	*n* = 19,169	*n* = 17,461	*n* = 1708	*n* = 1035	*n* = 657
**Maternal Age**					
** <25**	**26.9%**	26.7%	29.1%	33.2%	22.7%
** 25–34**	**55.3%**	55.5%	53.0%	51.5%	55.4%
** ≥35**	**17.8%**	17.8%	17.9%	15.5%	21.9%
**Maternal Race**					
** White Non-Hispanic**	**28.2%**	29.0%	19.4%	21.0%	17.0%
** Black Non-Hispanic**	**52.6%**	51.4%	64.4%	61.2%	69.7%
** Asian**	**6.3%**	6.4%	4.7%	6.3%	2.1%
** Hispanic**	**8.4%**	8.5%	7.6%	7.3%	7.6%
** Mixed/Other/Unknown**	**4.2%**	4.6%	3.9%	4.3%	3.5%
**BMI ^1^**					
** <25**	**49.5%**	50.2%	42.3%	49.2%	32.4%
** 25–<30**	**23.5%**	23.5%	23.3%	23.2%	23.8%
** ≥30**	**27.0%**	26.3%	34.4%	27.7%	43.8%
**Private Insurance ^2^**	**45.1%**	46.1%	35.8%	36.3%	35.4%
**Nulliparity ^3^**	**44.5%**	44.8%	41.3%	41.1%	42.0%
**Census Tract: Percentage of Poverty**	**28.0 (14.7)**	27.7 (14.8)	30.35 (14.1)	29.9 (14.2)	30.9 (13.8)
**Hospital of Birth**					
** Pennsylvania Hospital**	**54.8%**	55.6%	47.2%	46.4%	47.5%
** Hospital of the University of Pennsylvania**	**45.2%**	44.4%	52.8%	53.6%	52.5%
**Air Pollutants**					
** NO_2_ (ppb)**	**17.2 (2.3)**	17.2 (2.3)	17.1 (2.6)	17.2 (2.6)	17.0 (2.6)
** Benzene (ppb)**	**0.22 (0.02)**	0.22 (0.02)	0.22 (0.02)	0.22 (0.02)	0.22 (0.02)

^1^ Missing values for BMI overall: 3484, term: 3152, PTB: 332, sPTB: 236, mPTB: 86. ^2^ Missing values for private insurance overall: 314, term: 271, PTB: 443, sPTB: 33, mPTB: 10. ^3^ Missing values for nulliparity overall: 238, term: 237, PTB: 1, sPTB: 0, mPTB: 0.

**Table 2 ijerph-19-10365-t002:** Unadjusted and adjusted ^a^ odds ratios (ORs) of a standard deviation increment increase in NO_2_ and benzene with preterm birth (PTB) outcomes, including spontaneous (sPTB) and medically indicated (mPTB).

	Birth Outcome: PTB OR (95% CI)	Birth Outcome: sPTB OR (95% CI)	Birth Outcome: mPTB OR (95% CI)
Overall			
Unadjusted Benzene	0.96 (0.91, 1.01)	1.00 (0.94, 1.06)	0.89 (0.82, 0.97) *
Adjusted Benzene	1.02 (0.96, 1.08)	1.05 (0.97, 1.13)	0.97 (0.89, 1.06)
Unadjusted NO_2_	0.95 (0.90, 0.99) *	0.98 (0.92, 1.04)	0.90 (0.84, 0.97) *
Adjusted NO_2_	0.975 (0.92, 1.03)	0.972 (0.90, 1.05)	0.97 (0.89, 1.06)
Maternal race/ethnicity: Asian NH			
Unadjusted Benzene	1.06 (0.844, 1.33)	1.01 (0.78, 1.30)	1.26 (0.745, 2.04)
Adjusted Benzene	1.05 (0.817, 1.34)	0.974 (0.735, 1.29)	1.36 (0.80, 2.30)
Unadjusted NO_2_	1.11 (0.878, 1.42)	1.16 (0.891, 1.52)	0.925 (0.564, 1.63)
Adjusted NO_2_	1.12 (0.857, 1.46)	1.20 (0.892, 1.62)	0.833 (0.47, 1.47)
Maternal race/ethnicity: Black NH			
Unadjusted Benzene	1.04 (0.98, 1.10)	1.05 (0.97, 1.14)	1.01 (0.92, 1.11)
Adjusted Benzene	1.01 (0.94, 1.09)	1.04 (0.94, 1.15)	0.97 (0.88, 1.08)
Unadjusted NO_2_	1.01 (0.95, 1.09)	1.03 (0.94, 1.13)	1.00 (0.90, 1.11)
Adjusted NO_2_	0.99 (0.92, 1.07)	0.99 (0.90, 1.10)	1.00 (0.893, 1.12)
Maternal race/ethnicity: Hispanic			
Unadjusted Benzene	0.986 (0.782, 1.23)	1.09 (0.82, 1.43)	0.856 (0.58, 1.22)
Adjusted Benzene	1.05 (0.816, 1.35)	1.19 (0.861, 1.64)	0.929 (0.623, 1.39)
Unadjusted NO_2_	0.953 (0.78, 1.17)	0.980 (0.759, 1.29)	0.930 (0.684, 1.30)
Adjusted NO_2_	0.905 (0.715, 1.15)	0.855 (0.625, 1.17)	0.966 (0.677, 1.38)
Maternal race/ethnicity: White NH			
Unadjusted Benzene	0.96 (0.85, 1.08)	1.03 (0.89, 1.19)	0.82 (0.66, 1.00)
Adjusted Benzene	0.99 (0.87, 1.13)	1.02 (0.88, 1.19)	0.94 (0.75, 1.18)
Unadjusted NO_2_	0.95 (0.86, 1.05)	0.99 (0.88, 1.13)	0.87 (0.75, 1.02)
Adjusted NO_2_	0.95 (0.85, 1.05)	0.983 (0.858, 1.13)	0.88 (0.74, 1.05)
Maternal race/ethnicity: other			
Unadjusted Benzene	1.05 (0.815, 1.34)	1.02 (0.753, 1.38)	1.10 (0.719, 1.65)
Adjusted Benzene	1.12 (0.852, 1.48)	1.22 (0.875, 1.70)	0.962 (0.594, 1.56)
Unadjusted NO_2_	1.15 (0.911, 1.46)	1.03 (0.785, 1.37)	1.47 (0.971, 2.31)
Adjusted NO_2_	1.11 (0.850, 1.45)	0.964 (0.698, 1.33)	1.58 (0.962, 2.58)

^a^ Adjusted models were logistic mixed models adjusting for age, race, BMI, nulliparity, insurance status and census tract-level percentage below poverty. Hospital and census tracts were included as random intercepts. Adjusted models mutually adjusted for the other pollutant. NH, non-Hispanic. * Results were statistically significant.

## Data Availability

Data may be requested from Burris, and, upon approval from the University of Pennsylvania, deidentified data may be made available to external investigators. The data are not publicly available due to their containing protected health information.

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
