# Peer review of "Benzene and NO2 Exposure during Pregnancy and Preterm Birth in Two Philadelphia Hospitals, 2013–2017"

_ijerph, 2022, doi:10.3390/ijerph191610365_

Round 1

Reviewer 1 Report

I appreciate  the work that went into the analysis. this is an interesting piece but one that is too narrowly focused in my opinion. There are many confounding factors that contribute individually and collectively to pre-term birth. NO2 and benzene, as toxic as they are, are perhaps just a small part of the problem. You make no mention about length of exposure (how long had the pregnant woman lived in the area of exposure?), a more in depth discussion about the impact weight (BMI), personal lifestyle choices (smoking, alcohol drinking, diet, drug abuse, etc), prenatal care, impact of insurance status, SES, and other similar factors have on birth outcome is necessary. It is unrealistic to expect that exposure (and we dont know for how long based on what was presented in the paper) to NO2 and benzene would explain much of pre-term birth.

Reviewer 2 Report

This is an interesting investigation of environmental pollution and preterm labor. 

However, even though the authors had a large sample size, 18, 616 births, the use of only birth occurrence data without any information about preconception or trimester of pregnancy pollutant exposer is too crude of an indicator to identify any causative association.

The discussion and conclusion should be revised to expand on explanations for the author's finding of no association between environmental pollution and preterm birth.

Page 2 Line 50:  Young maternal age is not only associated with sPTB but is also associated with mPTD due to an increased risk of  preeclampsia for individuals at the extremes of reproductive age .  (Matern Child Health J. 2015 Jun; 19(6): 1202–1211.  doi: 10.1007/s10995-014-1624-7PMCID: PMC4418963 NIHMSID: NIHMS639929 PMID: 25366100

Round 2

Reviewer 1 Report

Thank you for trying to tease out other factors that potentially could contribute to pregnancy outcome in Philadelphia. Sadly, you do not have sufficient data on many of these factors (some if not most more important contributors to birth outcome than benzene an NO2). You might have had "better luck" studying birth outcomes in areas where fracking is prevalent, for example. But, again, many factors individually and collectively contribute to adverse birth outcomes.
